# Effect of Oxidation Time on the Properties of Cellulose Nanocrystals Prepared from Balsa and Kapok Fibers Using Ammonium Persulfate

**DOI:** 10.3390/polym13111894

**Published:** 2021-06-07

**Authors:** Marwanto Marwanto, Muhammad Iqbal Maulana, Fauzi Febrianto, Nyoman Jaya Wistara, Siti Nikmatin, Nanang Masruchin, Lukmanul Hakim Zaini, Seung-Hwan Lee, Nam-Hun Kim

**Affiliations:** 1Department of Forest Products, Faculty of Forestry and Environment, IPB University (Bogor Agricultural University), Bogor 16680, Indonesia; marwanto@workmail.com (M.M.); iqbal_2017@apps.ipb.ac.id (M.I.M.); nyomanwis@gmail.com (N.J.W.); lukmanhz@apps.ipb.ac.id (L.H.Z.); 2Department of Physics, Faculty of Mathematics and Natural Sciences, IPB University (Bogor Agricultural University), Bogor 16680, Indonesia; snikmatin@yahoo.com; 3Research Center for Biomaterials, Indonesian Institute of Sciences (LIPI), Cibinong 16911, Indonesia; masruchin@biomaterial.lipi.go.id; 4Department of Forest Biomaterial Engineering, College of Forest and Environmental Sciences, Kangwon National University, Chuncheon 24341, Korea; lshyhk@kangwon.ac.kr

**Keywords:** ammonium persulfate, balsa, cellulose nanocrystal, delignification, kapok

## Abstract

This study aimed to evaluate the effect of ammonium persulfate’s (APS) oxidation time on the characteristics of the cellulose nanocrystals (CNCs) of balsa and kapok fibers after delignification pretreatment with sodium chlorite/acetic acid. This two-step method is important for increasing the zeta potential value and achieving higher thermal stability. The fibers were partially delignified using acidified sodium chlorite for four cycles, followed by APS oxidation at 60 °C for 8, 12, and 16 h. The isolated CNCs with a rod-like structure showed an average diameter in the range of 5.5–12.6 nm and an aspect ratio of 14.7–28.2. Increasing the reaction time resulted in a gradual reduction in the CNC dimensions. The higher surface charge of the balsa and kapok CNCs was observed at a longer oxidation time. The CNCs prepared from kapok had the highest colloid stability after oxidation for 16 h (−62.27 mV). The CNCs with higher crystallinity had longer oxidation times. Thermogravimetric analysis revealed that the CNCs with a higher thermal stability had longer oxidation times. All of the parameters were influenced by the oxidation time. This study indicates that APS oxidation for 8–16 h can produce CNCs from delignified balsa and kapok with satisfactory zeta potential values and thermal stabilities.

## 1. Introduction

Indonesia has a rich biodiversity of tree species, especially hardwoods, and has developed kapok and balsa plantations for the timber industry. These plantations also produce fibrous fruits from both species as by-products. Indonesia is the biggest exporter of kapok commodities [1]. However, balsa fibers from the fruits are still unutilized. Balsa and kapok fruit fibers have potential as raw materials for cellulose-based products. Balsa and kapok fruit fibers are hollow fibers with a high cellulose and low lignin content [1]. Our previous research revealed that balsa and kapok fibers could be transformed into nanocellulose through oxidation [2].

Nanocelluloses are classified based on their morphology, such as cellulose nanofibrils and cellulose nanocrystals (CNCs) [3]. CNCs have beneficial properties such as a high elastic modulus, low thermal expansion, high aspect ratio, large specific surface area, non-abrasive nature, nontoxicity, and surface charge [4]. In general, CNCs are produced by acid hydrolysis using concentrated sulfuric acid. This method generates nanocellulose with high crystallinity and sulfate group-modified surfaces [5]. Moreover, CNCs obtained through hydrolysis can also be produced with varying combinations of acids [6,7,8,9,10], organic acids [11], and solid acids [12]. However, CNCs obtained through acid hydrolysis have a low thermal stability [5], limiting their applications because of the presence of a sulfate group on its surface [13].

Another study has been conducted to produce CNCs with a higher thermal stability, using the ammonium persulfate (APS) oxidation method. The APS oxidation method produces nanocellulose with a –COOH surface charge [14]. This method produces CNCs with a higher thermal stability than conventional hydrolysis with sulfuric acid [15,16]. However, compared to other methods (acid hydrolysis and TEMPO oxidation), the APS oxidation method produces a lower surface charge [17]. High thermal stability and high surface charge are specific characteristics of nanocellulose. They are required for expansion in many applications [5,18,19,20,21,22,23]. The surface charge of nanocellulose is described by the zeta potential and is formed during the isolation or surface modification process.

Previous research reported that the pretreatment of raw materials improved the zeta potential [17,24]. Moreover, the zeta potential was also affected by the oxidation time. A longer oxidation time results in a higher zeta potential [24]. Another pretreatment process used for impurity removal is delignification with sodium chlorite/acetic acid [25]. Sodium chlorite/acetic acid delignification can selectively remove lignin. It generates fibers with more roughness and porosity [26]. It is important to increase the penetration of the APS solution into the fibers.

Currently, there is no information on combining delignification with sodium chlorite/acetic acid (SC/AA) and APS oxidation methods to produce CNCs from a single fiber. This study aims to determine the effect of APS oxidation time after SC/AA delignification on the characteristics of balsa and kapok CNCs, especially their thermal stability and surface charge. This information provides a deeper understanding of the characteristics of balsa and kapok CNCs and their potential applications.

## 2. Materials and Methods

### 2.1. Materials

Kapok fibers (38.09% α-cellulose, 14.10% lignin, and 45.64% hemicellulose) and balsa fibers (44.62% α-cellulose, 16.60% lignin, and 37.35% hemicellulose) [1] were obtained from the plant collection of IPB University, Bogor, Indonesia. The balsa and kapok fibers were removed from their pods, separated from their seeds, and dried at room temperature. The fibers were milled with a laboratory cutting mill to prepare a 40–60 mesh fiber powder. The sodium chlorite (NaClO_2_) at 25% solution, acetic acid (CH_3_COOH), APS ((NH_4_)_2_S_2_O_8_) at 98% purity, and sodium hydroxide (NaOH) (purchased from Merck KGaA, Darmstadt, Germany) used were of reagent grade.

### 2.2. Delignification Process

Delignification was performed by adding the milled balsa or kapok fibers (3 g) to distilled water (300 mL) (fiber: solution ratio of 1:100 (m:v)) and kept in a water bath at 70 °C. Delignification was initiated by adding sodium chlorite (3 mL) and acetic acid (0.3 mL) to the suspension. The reaction was performed continuously for 1 h. The same amounts of sodium chlorite and acetic acid were added every hour, and the process was repeated for four cycles. The residue was filtered and washed with distilled water several times until the pH became neutral. The residue was then dried in an oven at 40 °C for 48 h.

### 2.3. Isolation of CNCs by APS Oxidation

The CNCs were prepared by APS oxidation following the procedure of Oun and Rhim [27] with modifications. A total of 2 g of delignified balsa or kapok fibers were added to 200 mL (fiber: solution ratio of 1:100 (m:v)) of 1 M APS in a 250 mL glass beaker. The mixtures were heated at 60 °C on a hot plate and stirred vigorously for 8, 12, or 16 h. The suspensions were centrifuged (high-speed refrigerated centrifuge with rotor type NA-8 and a capacity of 50 mL/12 tubes Suprema21, Tomy Kogyo Co. Ltd., Tokyo, Japan) at 10,000 rpm for 10 min to separate the APS solution. The obtained CNCs were then transferred to distilled water. The centrifuging and transfer processes were repeated until the pH of the suspensions reached 4.0. Sodium hydroxide (1 M) was added to the suspensions until the pH increased to approximately 7, followed by sonication for 30 min at 40% amplitude. The obtained CNC suspensions were kept in a cooler at 4 °C before characterization. The isolation process of the CNCs using the APS oxidation method is shown in Figure 1.

### 2.4. Transmission Electron Microscopy (TEM)

A drop of a diluted CNC suspension was dyed negatively with 0.5% aqueous uranyl acetate and placed onto a carbon-covered Cu framework, and then allowed to dry in a desiccator at room temperature. The size and shape of the CNCs were examined using TEM (FEI Tecnai G2 20 S-Twin, FEI, Eindhoven, The Netherlands) at 200 kV. The dimensions of the CNCs, including length and width, were determined using ImageJ software 1.53e (NIH, Bethesda, MD, USA). Each dimension was measured using 30 replicates. The experiment had a completely randomized design with one factor, oxidation time, at three levels (8, 12, and 16 h). The data were statistically evaluated by analysis of variance (ANOVA) using IBM SPSS Statistics 25.0 (IBM Co, Armonk, NY, USA). Furthermore, Duncan’s multiple range test was performed when there was a significant influence.

### 2.5. Zeta Potential (ZP) Analysis

The zeta potential of the CNCs was determined using an SZ-100 (Nanopartica series instrument, Horiba Scientific, Kyoto, Japan). Three samples were prepared from a stock suspension of 2.5 mg/mL and diluted to 0.01 wt% with distilled water. The zeta potential was tested in triplicates at 25 °C.

### 2.6. Functional Group Analysis

Functional group changes in the CNCs were analyzed using Fourier transform infrared spectroscopy (FTIR-UATR Perkin Elmer Spectrum Two, PerkinElmer Inc., Waltham, MA, USA). The analysis was conducted in the wavenumber range of 4000–400 cm^−1^ with 16 scans at 4 cm^−1^ resolution and a data interval of 1 cm^−1^.

### 2.7. Crystallinity Index (CI)

X-ray diffraction (XRD) analyses of each sample were performed on a Shimadzu XRD-7000 MaximaX instrument (Kyoto, Japan) with CuKα radiation (λ = 0.1542 nm) in the range of 2θ = 10–35, and at a speed of 4.8°/min. The following empirical Equation (1) derived from [28] was used to calculate CI:(1)$$\mathrm{CI}=\frac{I_{200}-I_{am}}{I_{200}}\;\times \;100$$
where *I*_200_ is the peak intensity corresponding to the crystalline region, and *I_am_* denotes the peak intensity of the amorphous fraction.

### 2.8. Thermal-Stability Testing

The thermal stability of the samples was measured using thermogravimetric analysis (TGA; STA7300 instrument, Hitachi, Tokyo, Japan). A CNC sample was heated from 27 °C to 600 °C at a heating rate of 10 °C/min. Measurements were performed in a nitrogen atmosphere at a gas flow rate of 10 mL/min.

## 3. Results and Discussion

### 3.1. Morphology and Particle Size Distribution of the CNCs in Suspensions

The appearance of the CNC suspensions is shown in Figure 2. The combination of the delignification and APS oxidation methods produced CNCs in a gel form. The gel-form of the CNCs indicated that the CNCs had higher surface charges on their surfaces [29]. The appearance of the CNCs was affected by the raw materials and the oxidation time. A longer oxidation time increases the transparency of the CNC suspension. The appearances of the balsa and kapok CNC suspensions were slightly different. In addition, the CNCs from the kapok fibers tended to be more transparent than those from the balsa. The high transparency of the CNCs was affected by the low wavelength CNCs of kapok [30].

The balsa and kapok CNCs show rod-like shapes (Figure 3). This confirms that the two-stage process performed with the APS oxidation method results in the extraction of individual CNCs from the fibers. The dimensions of the balsa and kapok CNCs depend on the raw material and oxidation time (Table 1). According to Rashid and Dutta [31], the diameter of CNCs is affected by the width of the cell wall of the raw material. Kapok fibers produce CNCs with smaller diameters than balsa fibers. This is probably due to kapok fibers having a smaller cell wall width than balsa fibers. The previous report shows that balsa and kapok fibers have a fiber cell wall width of 2.40 and 1.34 μm, respectively [1].

In this study, the nanostructures were produced with a shorter time (8 h) than the general APS oxidation process (16 h) [13]. Pretreatment shortens the oxidation time of the APS oxidation method. Pretreatment using sodium chlorite/acetic acid can reduce the fiber impurities and results in a rougher fiber surface [26]. Increasing the surface roughness of fibers increases their porosity in an aqueous medium [32]. This allows APS to penetrate the treated fibers more easily than the untreated fibers. A longer oxidation time increases the imbibition height and capillary mass of the fibers [33]. This accelerates the nanocellulose isolation process.

The average diameter of balsa and kapok CNCs after 8 h of oxidation time was 12.64 and 8.77 nm, respectively. At the same oxidation time, the average diameter of the kapok CNCs was lower than that of the balsa CNCs. Hence, the diameter of the CNCs prepared using this method was lower than that obtained with the acid hydrolysis method (16 nm) [34]. The dimensions of both CNCs significantly decreased (*p* < 0.01) with longer oxidation times. A longer oxidation time produced CNCs with a large surface area, which was due to a reduction in the dimensions (length and width) of the CNCs that was caused by the degraded and removed amorphous regions in cellulose during the APS oxidation process [35]. These results are in line with those of previous reports [24,35]. This phenomenon is also supported by the crystallinity index (CI). The CI increased owing to the longer oxidation time. The aspect ratios decreased gradually with the increasing oxidation time and produced a nanoparticle-like form. Nanoparticle-like morphological characteristics occurred after 16 h of APS oxidation. Both of the CNCs had lengths and diameters smaller than 100 nm.

### 3.2. Zeta Potential

The zeta potential reflects the colloidal stability of CNCs, which is affected by particle dispersion or agglomeration that is caused by electrostatic repulsive forces. The higher absolute value of the ZP generated a more stable suspension of the CNCs [35]. A higher ZP value resulted in a significant repulsive force. In addition, the repulsive forces on the CNCs prevented agglomeration. The ZP of the CNCs ranged from −10.23 to −62.27 mV, depending on the oxidation time and raw materials (Table 2). The ZP of the CNCs increased with the increasing oxidation time. The highest absolute ZP was achieved after 16 h of oxidation (62.27 mV) in the kapok CNCs, indicating an increase in the carboxyl group content after the oxidation process with the APS oxidation method. The primary hydroxyl groups on the cellulose surface were converted to carboxyl groups after the APS oxidation method [30]. The ZP value after 16 h oxidation time in this study was higher than that previously reported with the APS oxidation method and other methods (Table 2). This could be affected by performing the delignification process before the APS oxidation method in the fibers. The higher ZP value produced strong electrostatic repulsive forces between the CNCs, reduced the light scattering of the CNCs, and hindered their aggregation [31]. Generally, a stable suspension of nanocellulose shows a zeta potential value lower than −25.0 mV [16].

### 3.3. Functional Group Analysis

The untreated balsa and kapok fibers showed differences in the absorption bands measured at 1430–1250, 1610, 1750, and 2850 cm^−1^ due to the different amounts of extractive, hemicellulose, and lignin [37] (Figure 4). The balsa and kapok fibers showed absorbance bands in the spectra at 1595, 1508, and 1230 cm^−1^, assigned to C=O and C=C aromatic skeletal vibrations, indicating the presence of lignin and hemicellulose [38,39]. These bands were not visible in the CNC spectra after combining the delignification and oxidation processes. This combined process can remove non-cellulosic components in balsa and kapok fibers.

The CNCs of balsa and kapok fibers exhibited typical peaks at broad O–H stretching (3600–3100 cm^−1^), C–H stretches (3000–2800 cm^−1^), C–H bends (1450–1300 cm^−1^), and C–O stretching (1170–1050 cm^−1^) [38]. The CNCs showed more intense cellulose peaks at ~1159, 1106, and 1055 cm^−1^ [38]. The intense peaks corresponding to the vibrations of cellulose indicates the high purity of the CNCs after the combined process [40]. The balsa and kapok CNCs exhibited a peak of C=O stretching vibration at 1733 cm^−1^. The carboxylic group is a typical characteristic of CNCs prepared using the APS oxidation method [13].

### 3.4. Crystallinity Index

CNCs show a pattern typical of cellulose I (Figure 5). The delignification treatment followed by oxidation did not change this pattern. The CI increased with oxidation time. These results indicate that delignification and APS oxidation did not damage the crystalline region [41]. The same phenomenon was observed in a previous study [35,42]. The cellulose I patterns showed major diffraction peaks at 2θ of approximately 15.1° (I 1 −1 0), 16.5° (I 1 1 0), 20.4° (I 0 1 2), and 22.6° (I 2 0 0) [35]. The single crystallinity peaks of the balsa CNCs resulting from 8, 12, and 16 h of oxidation were 22.09° (I 2 0 0), 21.86° (I 2 0 0), and 21.79° (I 2 0 0), respectively. Meanwhile, the crystallinity peaks of the kapok CNCs resulting from 18, 12, and 16 h of oxidation were 21.93° (I 2 0 0), 22.12° (I 2 0 0), and 22.41° (I 2 0 0), respectively.

The combined method of delignification and APS oxidation generated a higher CI of CNCs than the raw materials. The CI of the balsa and kapok CNCs increased by 51.24–83.60 and 55.82–70.29%, respectively (Table 3). The increase in the CI of the CNCs was due to the removal of the impurities and amorphous regions of the cellulose during the delignification and APS oxidation processes [27]. These results are consistent with the FTIR analyses. The CI of the balsa CNCs was lower than that of the kapok CNCs. Leung et al. [13] reported that raw materials affect the degree of crystallinity. The longer oxidation time also generated a higher CI of the balsa and kapok CNCs. This is probably because the successive hydrolysis of the amorphous domains results in a higher amount of crystalline domains of the CNCs [35]. The balsa and kapok CNCs in this study have a higher CI compared to the CNCs from the APS oxidation method without delignification at the same concentration [2].

### 3.5. Thermal Stability

The thermal stability of the CNCs is lower than that of the original fibers (Figure 6). All of the samples decrease in weight at a low temperature (<60–120 °C), suggesting the evaporation of loosely bound moisture [27]. The *T*_onset_ decomposition temperatures of the balsa and kapok fibers are 263.80 and 254.49 °C, respectively (Table 4). This is caused by the differences in the chemical composition and morphology of the raw materials. However, the *T*_onset_ of the CNCs was below 250 °C. The lower thermal degradation temperature of the CNCs is due to the smaller size of the CNCs. The decreased degradation temperature of the CNCs may be related to their greater surface area compared to the original fibers [43]. Moreover, the decreased degradation temperature may be due to the disruption of hydrogen bonding in the original cellulose upon the addition of the carboxyl group [16].

The TGA results show that the thermal stability of the kapok fiber CNCs is lower than that of the balsa at the same oxidation time. This is probably due to the smaller size of the CNCs in kapok fibers than in the balsa CNCs. The thermal stability of CNCs is affected by several factors, such as dimensions, specific surface area, and molecular weight [24]. The thermal stability of the balsa and kapok CNCs was also influenced by the oxidation time. The CNCs isolated for 16 h had a higher thermal stability than the rest. The differences in the thermal decomposition profiles of the CNCs could be affected by varying amounts of surface charge [35] and degree of crystallinity [14]. This is confirmed by the higher zeta potential value and degree of crystallinity with longer oxidation times. The presence of −COOH groups on the CNCs surface can increase the thermal stability of the CNCs [5]. The *T*_onset_ of the CNCs in this study was higher than that of the CNCs obtained from sulfuric acid hydrolysis [15,16]. The higher *T*_onset_ of the CNCs derived from the APS oxidation method was due to the presence of carboxyl groups [16].

The two-step isolation of CNCs using the APS oxidation method in balsa and kapok fibers had a higher thermal stability than the CNCs from balsa and kapok reported in a previous study [2]. The CNCs from untreated balsa and kapok fibers oxidized by 1 M APS for 16 h thermally degrade at 191.5–221.8 and 228.2–264.97 °C, respectively. Conversely, in the present study, the balsa and kapok CNCs fabricated by oxidation with 1 M APS for 8 h show the following higher values: 219.71–277.87 and 215.35–261.39 °C, respectively. The combination of the delignification and APS oxidation processes increased the thermal stability of the balsa and kapok CNCs. This indicates that pretreatment could improve the thermal stability of CNCs. The higher thermal stability of CNCs after longer oxidation times may be explained by the higher colloidal stability and higher crystallinity degree.

## 4. Conclusions

The combination of delignification and APS oxidation produced balsa and kapok CNCs with a high zeta potential and a high thermal stability. CNCs have a rod-like structure and gel form. The CNCs of balsa and kapok showed an average diameter of 6.52–12.64 and 5.82–8.77 nm. Longer oxidation times resulted in smaller diameters and shorter CNCs. In addition, the zeta potential and CI of the balsa and kapok CNCs increased with the increasing oxidation time. Longer oxidation times resulted in the higher thermal stability of the CNCs. These characteristics can expand the use of CNCs in many applications, such as nanofillers, biomedicine, drug delivery, flocculants, and adsorbents.

## Figures and Tables

**Figure 1 polymers-13-01894-f001:**
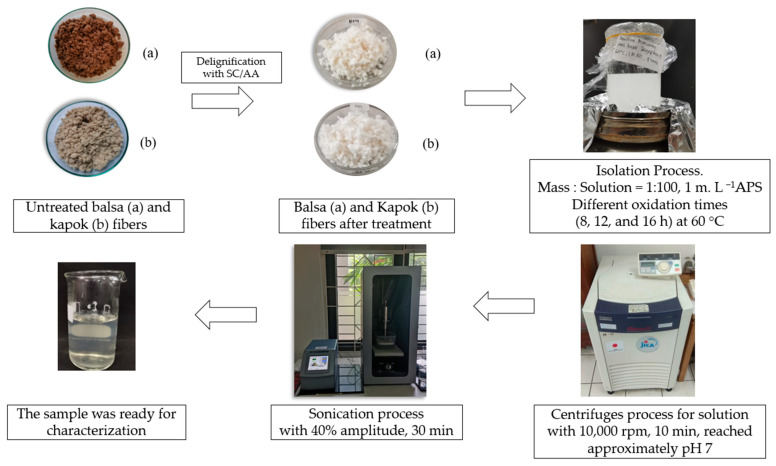
Isolation process of CNCs with the APS oxidation method.

**Figure 2 polymers-13-01894-f002:**
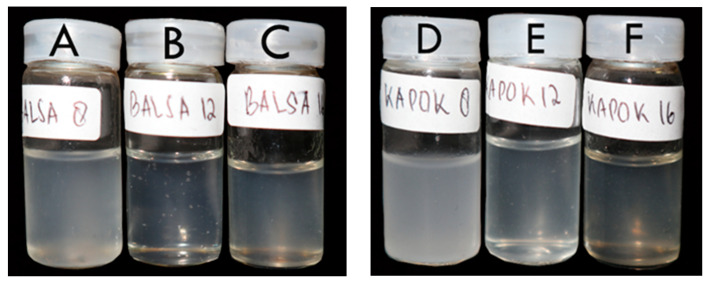
Appearance of CNC suspensions after oxidation using the APS method with different oxidation times. (**A**) Balsa, 8 h oxidation; (**B**) balsa, 12 h oxidation; (**C**) balsa, 16 h oxidation; (**D**) kapok, 8 h oxidation; (**E**) kapok, 12 h oxidation, and (**F**) kapok, 16 h oxidation.

**Figure 3 polymers-13-01894-f003:**
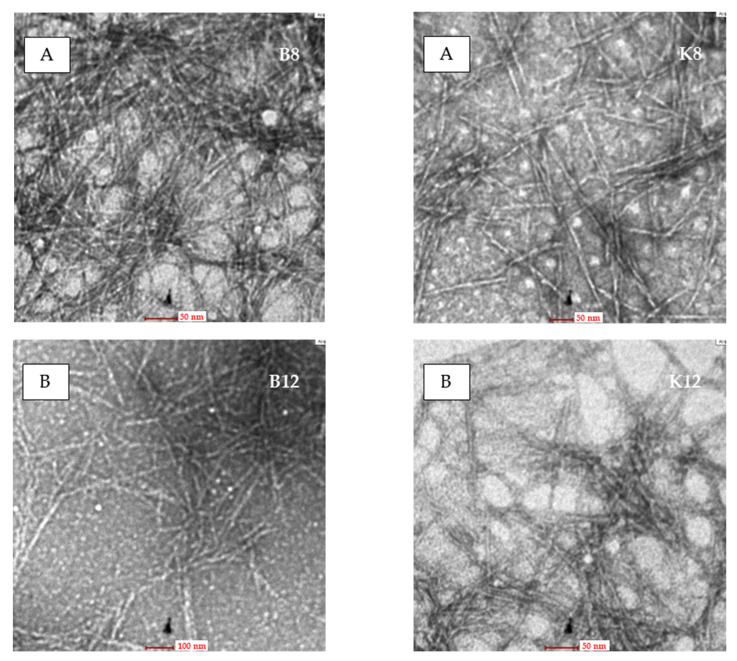
Transmission electron microscopy images of the rod-like CNCs obtained by APS oxidation of balsa fiber (**left**) and kapok fiber (**right**) for different oxidation times: (**A**) 8 h, (**B**) 12 h, and (**C**) 16 h.

**Figure 4 polymers-13-01894-f004:**
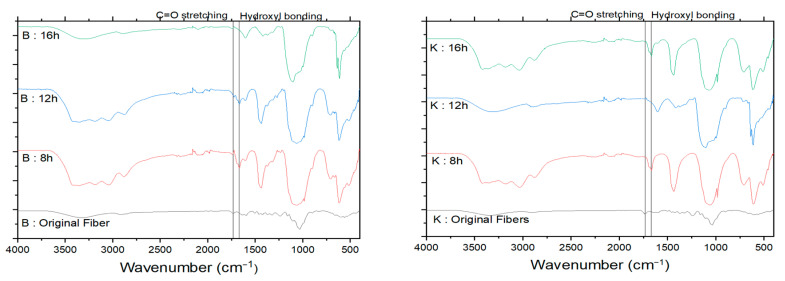
FTIR spectra of balsa (**left**) and kapok (**right**) fibers and the resultant CNCs.

**Figure 5 polymers-13-01894-f005:**
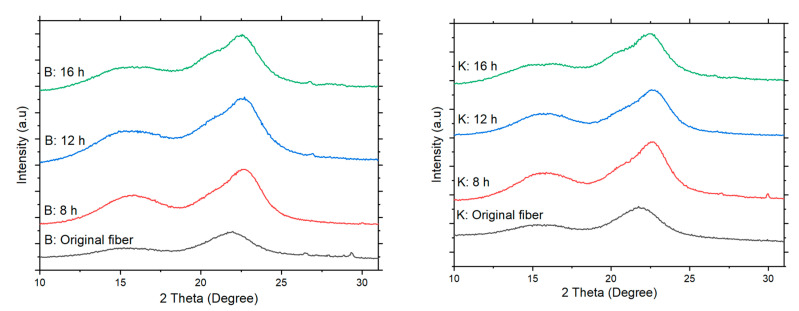
X-ray diffraction (XRD) patterns of the (**left**) balsa fiber and (**right**) kapok fiber and the resulting CNCs.

**Figure 6 polymers-13-01894-f006:**
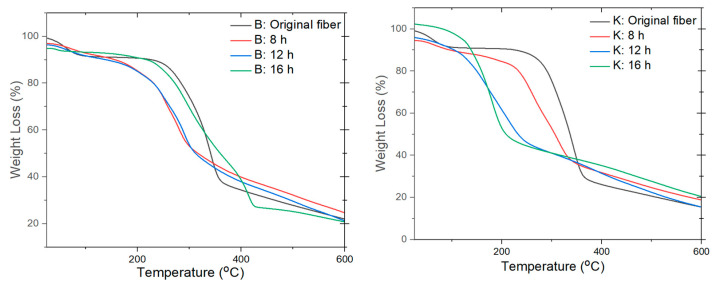
Thermogravimetric analysis (TGA) curves of the balsa fiber (**left**) and kapok fiber (**right**) and the resulting CNCs.

**Table 1 polymers-13-01894-t001:** Balsa and kapok fiber CNC size distributions.

Samples	Reaction Time (h)	Diameter (D, nm)			Length (L, nm)			Aspect Ratio (L/D)
		Min.	Max.	Ave. *	Min.	Max.	Ave. *	
CNCs from Balsa	8	7.35	19.48	12.64 (a0)	115.33	482.48	274.40 (a2)	21.72
	12	6.25	16.11	10.44 (b0)	182.57	397.60	261.58 (a2)	25.06
	16	3.93	11.25	6.52 (c0)	52.60	182.65	95.88 (b2)	14.72
CNCs from Kapok	8	4.62	13.86	8.77 (a1)	189.96	369.81	247.61 (a3)	28.23
	12	2.78	9.16	5.54 (b1)	93.27	186.91	136.36 (b3)	24.61
	16	3.14	8.26	5.82 (b1)	82.87	152.36	112.06 (c3)	19.24

* Different letters in the same column denote significant differences between treatments according to Duncan’s multiple range test with a 5% confidence interval.

**Table 2 polymers-13-01894-t002:** Zeta potentials (ZPs) of the CNCs from balsa and kapok fibers.

Sample	Reaction Time (h)/Methods	ZP/mV	Literature
CNCs from balsa	8	−29.93	In this study
	12	−44.23	In this study
	16	−51.43	In this study
CNCs from kapok	8	−10.23	In this study
	12	−61.87	In this study
	16	−62.27	In this study
CNCs from cotton	APS oxidation methods	−50.60	[14]
CNCs from MCC	APS oxidation methods	−46.90	[14]
CNCs from jute fiber	APS oxidation methods	−40.00	[24]
CNCs from denim waste	APS oxidation methods	−3.53	[17]
CNCs from lemon pulp	APS oxidation methods	−31.27	[16]
CNCs from lemon pulp	Sulfuric acid (Acid hydrolysis)	−40.27	[16]
CNCs from lemon pulp	TEMPO Oxidation	−55.67	[16]
CNCs from cotton pulp	Phosphoric acid (Acid hydrolysis)	−17.03	[36]

**Table 3 polymers-13-01894-t003:** Crystallinity indices (CIs) of the balsa and kapok fibers and of the resulting CNCs.

Sample		CI, °	
		Balsa	Kapok
Original fiber		31.40	35.65
CNCs	8 h	47.49	55.55
	12 h	51.00	52.50
	16 h	57.65	60.71

**Table 4 polymers-13-01894-t004:** Decomposition onset temperature and maximum thermal degradation temperature (*T*_onset_ and *T*_max_) of untreated balsa and kapok fibers and their resulting CNCs.

Sample	Reaction Time (h)	Temperature (°C)	
		*T* _onset_	*T* _max_
Balsa	Original fiber	263.80	342.64
	8	219.71	277.87
	12	240.73	289.91
	16	241.33	291.57
Kapok	Original fiber	254.49	344.80
	8	215.35	261.39
	12	216.97	273.93
	16	233.45	280.64

## Data Availability

Not applicable.

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
