# Peer review of "Effect of Oxidation Time on the Properties of Cellulose Nanocrystals Prepared from Balsa and Kapok Fibers Using Ammonium Persulfate"

_polymers, 2021, doi:10.3390/polym13111894_

Round 1

Reviewer 1 Report

Nanocellulose is an innovative material obtained from renewable and abundant bioresources. Nanocelluloses can be classified based on their morphology, for example, as cellulose nanofibrils and cellulose nanocrystals (CNCs). CNCs have beneficial properties such as high elastic modulus, low thermal expansion, a high aspect ratio, large specific surface area, a nonabrasive nature, and nontoxicity. These properties make CNCs suitable for many applications, including oil-water separation aerogels, protein immobilization, antimicrobial packaging, drug delivery, nanocomposite reinforcement, and optically transparent films. In this article, the authors aimed to evaluate the preparation of cellulose nanocrystals (CNCs) from balsa and kapok fiber by ammonium persulfate oxidation after delignification, with different oxidation times. Morphological characteristics and surface properties were found to be affected by oxidation time. Although the topic in this work was interesting, the presentation in this manuscript was very poor. This manuscript should be rejected for published in Polymers. However, if the authors are willing to make the substantial revisions according to my comments, I would be glad to re-review this manuscript. Here are my detailed comments:

  1. The authors need to reorganize the current introduction, which normally consists of three parts at least: background, literature review, brief of the proposed work. The current one is nothing but a literature review. Why their work is important comparing to previous reports? I think this is essential to keep the interest of the reader.
  2. In Fig. 6, the authors should give the explanations for the difference of data collected from different sources.
  3. Materials and Methods part, Although the results look “making sense”, the current form reads like a simple lab report. The authors should dig deeper in the results by presenting some in-depth discussion.
  4. The isolated CNCs showed the average diameter in the range of 5.54–12.64 nm, leading to an aspect ratio of 14.72–28.22 with a high surface charge. With the increasing reaction time, a gradual reduction in the size of the CNCs with rod-like structure was observed. The authors should give some explanation on above conclusions and data.
  5. The present work mainly focuses on lab work. It does not necessarily imply that the theoretic work (modeling) is not important. The authors omit this part during the current literature review, which should include a brief review of the theoretic work after revision. In the theoretic perspective, fractal theory is a very important tool, which can be used to investigate the surface morphology and fiber diameter of fibers (see [A fractal model for capillary flow through a single tortuous capillary with roughened surfaces in fibrous porous media, Fractals, 2021, 29(1):2150017; Fractals, 2019, 27(7): 1950116]). Authors should introduce some related knowledge to readers. I think this is essential to keep the interest of the reader.
  6. Please, expand the conclusions in relation to the specific goals and the future work.

Author Response

Please find the response attached.

Reviewer 2 Report

Marwanto et al. demonstrated the preparation of cellulose nanocrystals (CNCs) from Balsa and Kapok fiber using ammonium persulfate (APS) oxidation and investigated the effect of oxidation time on the properties of the obtained CNCs. However, the strategy using APS oxidation for the production of CNCs has been already reported, and thus it is routine work. The only novelty of this work is the cellulose source, also highlighted by the author. In fact, CNCs can be produced via chemical pretreatment and APS oxidation from every natural fiber and agricultural biomass on the earth and keep you busy for the rest of your life without adding anything of real value. Because of lack of novelty, I do not recommend the manuscript to be published in a highly prestigious journal like Polymers. More issues are listed below for the authors’ reference.

  1. The manuscript is written in poor English with lots of word use, grammar, and stylistic issues, which is not acceptable for publication in its current form.
  2. Although the authors did necessary characterization on the prepared CNCs, the discussions are kind of superficial, deeper explanation combined with comparison with other reported methods (e.g. the traditional sulfuric acid hydrolysis) in the literature should be added.
  3. I would suggest that the author come up with a new application for the produced CNCs which shows the real value for using this material rather than just extraction of it and just referring to other publications to introduce various potential applications of such materials.

Author Response

Please find the response attached.

Reviewer 3 Report

The authors submitted a manuscript on effect of oxidation time on the properties of cellulose nanocrystals prepared from balsa or kapok fiber using ammonium persulfate. The topic is within the scope of the Journal and the results are up-to-date. However, there are some major issues that should be properly addressed by the authors before it can be accepted for publication.
As a first comment, the English language and the spelling should be carefully revised because it does not completely meet the requirements of a scientific publication.
1.     Introduction: I would suggest the authors to highlight  the novelty of the present research, why it is needed in the first place and how it is supposed to add significantly to the available literature in the field;
2.     As regards the experimental section, some details and clarifications should be added;
3.     As a general comment, the discussion of results lacks a thorough analysis, it is mostly a description of what can be inferred from the plots.

Author Response

Please find the response attached.

Round 2

Reviewer 1 Report

Although the author provided detailed point-by-point response and made correction in the manuscript. The response is of no significance. The revised form simply ignore my comments made in the first round. This manuscript should be rejected for published in Polymers. However, if the authors are willing to make the substantial revisions according to my the first comments, I would be glad to re-review this manuscript.

Author Response

Dear reviewer

The responses to your comments are presented in the table below. In addition, please see the attachment for the revised manuscript. The revised sentences in the manuscript are presented with the track changes. We hope our response and revision can reach the requirement.

Comment:

Although the author provided detailed point-by-point response and made correction in the manuscript. The response is of no significance. The revised form simply ignore my comments made in the first round. This manuscript should be rejected for published in Polymers. However, if the authors are willing to make the substantial revisions according to my first comments, I would be glad to re-review this manuscript.

Revision:

Thank you for your kind comment. We revised the manuscript carefully according to your suggestion in the first round.

Comment:

The authors need to reorganize the current introduction, which normally consists of three parts at least: background, literature review, brief of the proposed work. The current one is nothing but a literature review. Why their work is important comparing to previous reports? I think this is essential to keep the interest of the reader.

Revision:

Thank you for your kind comment. We revised the introduction according to your suggestion. (Line 34-74).

Background (Lines 35-42)

Indonesia has high tree species biodiversity, especially hardwood species, and developed forest plantations for kapok and balsa trees for the lumber industries. These plantations also produce fiber fruit from both species as a by-product. Indonesia is the biggest exporter country of kapok commodity [1]. However, balsa fiber from its fruits is still unutilized. Balsa and kapok fruit fibers have the potential as raw materials for cellulose-based products. Balsa and kapok fruit fibers belong to single fibers with a hollow form with high cellulose and lower lignin contents [1]. Our previous research revealed that balsa and kapok fibers could be generated to nanocellulose by oxidation method [2].

Literature review (Lines 43-68)

Nanocelluloses are classified based on their morphology, such as cellulose nanofibrils and cellulose nanocrystals (CNCs) [3]. Cellulose nanocrystals have beneficial properties such as high elastic modulus, low thermal expansion, a high aspect ratio, large specific surface area, a nonabrasive nature, nontoxicity, and surface charge [4]. In general, CNCs were produced from the acid hydrolysis method using concentrated sulfuric acid. This method generates nanocellulose with high crystallinity and sulfate groups modified surface [5]. Moreover, the CNCs with hydrolysis process also can be produced with the varying combination of acid [6] [7] [8] [9] [10], organic acid [11], and solid acid [12]. However, CNCs from the acid hydrolysis process have low thermal stability [5], limiting their applications due to the presence of a sulfate group on its surface [13].

Another research has been conducted to produce CNCs with higher thermal stability using the ammonium persulfate (APS) oxidation method. APS oxidation method produces nanocellulose with -COOH surface charge [14]. This method produced CNCs with higher thermal stability than conventional hydrolysis with sulfuric acid [15] [16]. However, compared with other methods (acid hydrolysis and TEMPO oxidation), the APS oxidation method produced a lower surface charge [17].  High thermal stability and high surface charge are specific characteristics of nanocellulose. They were required for expansion in many applications [5] [18] [19] [20] [21] [22][23]. The surface charge of nanocellulose is described by zeta potential and formed during the isolation or surface modification process.

Previous research mentioned that the pretreatment of raw materials improved the zeta potential [17] [24]. Moreover, the zeta potential was also affected by oxidation time. The longer oxidation time produces higher zeta potential [24]. Another pretreatment process for impurities removal was delignification with sodium chlorite/acetic acid [25]. Sodium chlorite/acetic acid delignification was able to remove lignin selectively. It generates fiber more roughness and porous [26]. It is important for increasing the penetration of the APS solution on fibers.

Brief of proposed work (Lines 69-74)

Up to now, there is no information on combining delignification with sodium chlorite/acetic acid (SC/AA) and APS oxidation methods to produce CNCs from a single fiber. This study aims to determine the effect of APS oxidation time after SC/AA delignification on the characteristics of CNCs balsa and kapok, especially their thermal stability and surface charge. This information provides deeper understanding characteristics of CNCs balsa and kapok and their potential applications.

Comment: In Fig. 6, the authors should give the explanations for the difference of data collected from different sources.

Revision:

Thank you for your kind suggestion. We added several studies in the discussion section (lines 276-308)

Lines 276-308

The thermal stability of CNCs was lower than that of the original fibers (figure 6). All samples decreased in weight at a low temperature (<60–120 °C), suggesting evaporation of loosely bound moisture [27]. Tonset decomposition temperature of balsa and kapok fibers was 263.80 and 254.49 °C, respectively (Table 4). It is caused by differences in the chemical composition and morphology of the raw materials. However, Tonset of their CNCs was below 250 °C. The lower thermal degradation temperature of CNCs is due to the smaller size of CNCs. The decreased degradation temperature of CNCs may be related to the greater surface area than the original fibers [43]. Moreover, the decreased degradation temperature may be due to the disruption of hydrogen bonding in the original cellulose upon the addition of the carboxyl group [16].

The TGA results show that the thermal stability of CNCs of kapok fiber is lower than CNCs of balsa at the same oxidation time. It was probably due to CNCs of kapok fiber have a smaller size than the CNCs of balsa. The thermal stability of CNCs is affected by several factors, such as dimensions, specific surface area, and molecular weight [41]. The thermal stability of CNCs of balsa and kapok was also influenced by oxidation time. Cellulose nanocrystals isolated for 16 h oxidation time has higher thermal stability than other CNCs. The differences in thermal decomposition profiles of CNCs could be affected by varying amounts of surface charge and degree of crystallinity [27]. It is confirmed by the higher zeta potential value and degree of crystallinity by longer oxidation time. The presence of -COOH groups on the CNCs surface can increase the thermal stability of the CNCs [5]. The Tonset of CNCs in this study were higher than CNCs from sulfuric acid hydrolysis [15] [16]. The higher Tonset of CNCs from APS oxidation method was due to the presence of carboxyl groups [16].

The two-step isolation CNCs using the APS oxidation method in balsa and kapok fibers had higher thermal stability than CNCs from balsa and kapok reported in the previous study [2]. Cellulose nanocrystals from untreated balsa and kapok oxidized by 1 M APS for 16 h thermally degrade at 191.5–221.8 °C and 228.2–264.97 °C, respectively. By contrast, in the present study, the balsa and kapok CNCs fabricated by oxidation with 1 M APS for 8 h show higher values: 219.71–277.87 °C and 215.35–261.39 °C, respectively. Combining the delignification process and APS oxidation methods increased the thermal stability properties of CNCs from balsa and kapok. It means that pretreatment could improve the thermal stability of CNCs. Higher thermal stability of CNCs after longer oxidation may be explained by higher colloidal stability and a higher degree of crystallinity.

Comment: Materials and Methods part, Although the results look “making sense”, the current form reads like a simple lab report. The authors should dig deeper in the results by presenting some in-depth discussion.

Revision:

Thank you for your kind comment. We revised the research results with in-depth discussions in several sections presented in the manuscript based on your suggestions.

Line 144-218 (Morphology and Particle Size Distribution of the CNCs in Suspensions),

Line 203-218 (Zeta Potential),

Line 233-247 (Functional Group Analysis),

Line 250-270 (Crystallinity Index),

Line 276-308 (Thermal Stability).

We hope that your suggestions have been accommodated in this revision.

Comment:

The isolated CNCs showed the average diameter in the range of 5.54–12.64 nm, leading to an aspect ratio of 14.72–28.22 with a high surface charge. With the increasing reaction time, a gradual reduction in the size of the CNCs with rod-like structure was observed. The authors should give some explanation on above conclusions and data.

Revision:

Thank you for your suggestion. We revised lines 152-166 and 171-183 in the discussion section according to your comment

Lines 152-166

The balsa and kapok CNCs showed a rod-like shape (Figure 3). It confirms that the two-stage process with the APS oxidation method results in individual extraction of the CNCs of fibers. The dimensions of CNCs balsa and kapok depend on raw material and oxidation time (Table 1). According to Rashid and Dutta [31], the diameter of CNCs was affected by the width cell wall of raw material. Kapok fibers produce a smaller diameter of CNCs than balsa fibers. This probably due to kapok fibers having a lower width cell wall of fiber compared to balsa fibers. The previous report shows balsa and kapok fibers have a width cell wall fiber of 2.40 μm and 1.34 μm, respectively [1].

Lines 171-183

The average diameter of balsa and kapok CNCs after 8 h of oxidation time was 12.64 and 8.77 nm, respectively.  At the same oxidation time, CNCs kapok has an average diameter lower than CNCs balsa. Hence, the diameter of CNCs kapok with this method is lower than the acid hydrolysis method (16 nm) [34]. The dimensions of both CNCs have significantly decreased (P < 0.01) with longer oxidation time. The longer oxidation time produced CNCs with a large surface area, which is due to a reduction in the dimensions (length and width) of CNCs caused by degraded and removed amorphous regions in cellulose during the APS oxidation process [35]. These results were in line with the previous reports [35] [24]. This phenomenon is also supported by the crystallinity index (CI). The crystallinity index increased due to longer oxidation time.  The aspect ratios decreased gradually with increasing oxidation time and produced a nanoparticle-like form. Nanoparticle-like morphological characteristics occurred after 16 h APS oxidation time. Both CNCs had a length and diameter below 100 nm.

Comment:

The present work mainly focuses on lab work. It does not necessarily imply that the theoretic work (modeling) is not important. The authors omit this part during the current literature review, which should include a brief review of the theoretic work after revision. In the theoretic perspective, fractal theory is a very important tool, which can be used to investigate the surface morphology and fiber diameter of fibers (see [A fractal model for capillary flow through a single tortuous capillary with roughened surfaces in fibrous porous media, Fractals, 2021, 29(1):2150017; Fractals, 2019, 27(7): 1950116]). Authors should introduce some related knowledge to readers. I think this is essential to keep the interest of the reader

Revision:

Thank you for your suggestion.

We revised lines 64-68 in the introduction section and lines 160-166 in the discussion section according to your suggestion.

Lines 64-68

Another pretreatment process for impurities removal was delignification with sodium chlorite/acetic acid [25]. Sodium chlorite/acetic acid delignification was able to remove lignin selectively. It generates fiber more roughness and porous [26]. It is important for increasing the penetration of the APS solution on fibers.

Lines 160-166

In this study, the nanostructures produced with a shorter time (8h) than the general APS oxidation process (16 h) [13]. The pretreatment shortens the oxidation time in the APS oxidation method. The pretreatment using sodium chlorite/acetic acid can reduce fiber impurity and result in a rougher fiber surface [26]. The increasing surface roughness on fiber increases the porosity in an aqueous medium [32]. It allows for APS to be able to penetrate better than untreated fibers. The longer oxidation time increases the imbibition height and mass of the capillary on the fibers [33]. It accelerates the nanocellulose isolation process.

Comment:

Please, expand the conclusions in relation to the specific goals and the future work

Revision:

Thank you for your suggestion. We revised the conclusion according to your suggestion.

Lines 318-325

The combining delignification and APS oxidation method produced balsa and kapok CNCs with high zeta potential and high thermal stability. The cellulose nanocrystals have a rod-like structure and gel form. The cellulose nanocrystals of balsa and kapok showed an average diameter of 6.52–12.64 and 5.82–8.77 nm. Longer oxidation time resulted in a smaller diameter and shorter CNCs. In addition, the zeta potential and CI of balsa and kapok CNCs were increased by longer oxidation time. The longer oxidation time produces higher thermal stability of CNCs. Those characteristics can expand the use of CNCs in many applications such as nanofiller, biomedical, drug delivery, flocculant, or adsorbent

Reviewer 2 Report

The authors have shown good effort in addressing my previous comments. I would like to add the followings:

  1. English can be improved again.
  2. In the Introduction, please state clearly, how this study is different from the previous ones, what are the novel points and why it is needed to conduct this study. Do the same for the abstract, in the first two to three lines please give the main problem or question.
  3. A brief literature review focusing on the preparation methods of CNC should be added in the Introduction section, some of the recently developed methods such as organic acid hydrolysis (e.g., FeCl3 catalyzed formic acid or citric acid hydrolysis), mixed acid hydrolysis (e.g., H2SO4/Acetic acid, H2SO4/formic acid, H2SO4/oxalic acid), solid acid hydrolysis are suggested to be mentioned.
  4. As for the application of CNC, the following references are suggested to be added to make a complete literature review: Carbohydr. Polym., 2019, 209, 130-144; ACS Sustain. Chem. Eng., 2020, 8(20), 7536-7562. 

Author Response

Dear reviewer

The responses to your comments are presented in the table below. In addition, please see the attachment for the revised manuscript. The revised sentences in the manuscript are presented with the track changes. We hope our response and revision can reach the requirement.

Comment:

The authors have shown good effort in addressing my previous comments. I would like to add the followings:

Revision:

Thank you for your kind comment.

Comment:

English can be improved again.

Revision:

Thank you for your kind comment. We revised the manuscript carefully.

Comment:

In the Introduction, please state clearly, how this study is different from the previous ones, what are the novel points and why it is needed to conduct this study. Do the same for the abstract, in the first two to three lines please give the main problem or question.

Revision:

Thank you for your comments and suggestion. We revised the lines 17-20 in the abstract section and lines 69-74 in the introduction section.

Lines 17-20

This study aimed to evaluate the effect of ammonium persulfate (APS) oxidation time on cellulose nanocrystals (CNCs) characteristics of balsa and kapok fibers after delignification pretreatment with sodium chlorite/acetic acid. This two-step method was important to increase the zeta potential value and to achieve higher thermal stability. The fiber was partially delignified using acidified sodium chlorite for 4 cycles and followed by the APS oxidation method at 60 °C for 8, 12, and 16 hours. The isolated CNCs with a rod-like structure showed the average diameter in the range of 5.5–12.6 nm and the aspect ratio of 14.7–28.2. Increasing reaction time resulted in a gradual reduction in the CNCs dimension. The higher surface charge of balsa and kapok CNCs was found at a longer oxidation time. The CNCs prepared from kapok had the highest colloid stability after oxidation for 16 h:-62.27 mV. Cellulose nanocrystals with higher crystallinity had a longer oxidation time. Thermogravimetric analysis revealed that CNCs with higher thermal stability had a longer oxidation time. All the parameters were influenced by oxidation time. This study indicates that APS oxidation for 8–16 h can produce CNCs from delignified balsa and kapok with satisfied zeta potential value and thermal stability.

Brief of proposed work (Lines 69-74)

Up to now, there is no information on combining delignification with sodium chlorite/acetic acid (SC/AA) and APS oxidation methods to produce CNCs from a single fiber. This study aims to determine the effect of APS oxidation time after SC/AA delignification on the characteristics of CNCs balsa and kapok, especially their thermal stability and surface charge. This information provides deeper understanding characteristics of CNCs balsa and kapok and their potential applications.

Comment:

A brief literature review focusing on the preparation methods of CNC should be added in the Introduction section, some of the recently developed methods such as organic acid hydrolysis (e.g., FeCl3 catalyzed formic acid or citric acid hydrolysis), mixed acid hydrolysis (e.g., H2SO4/Acetic acid, H2SO4/formic acid, H2SO4/oxalic acid), solid acid hydrolysis are suggested to be mentioned.

Revision:

Thank you for your suggestion. We added lines 43-52 in the introduction section.

Lines 43-52

Nanocelluloses are classified based on their morphology, such as cellulose nanofibrils and cellulose nanocrystals (CNCs) [3]. Cellulose nanocrystals have beneficial properties such as high elastic modulus, low thermal expansion, a high aspect ratio, large specific surface area, a nonabrasive nature, nontoxicity, and surface charge [4]. In general, CNCs were produced from the acid hydrolysis method using concentrated sulfuric acid. This method generates nanocellulose with high crystallinity and sulfate groups modified surface [5]. Moreover, the CNCs with hydrolysis process also can be produced with the varying combination of acid [6] [7] [8] [9] [10], organic acid [11], and solid acid [12]. However, CNCs from the acid hydrolysis process have low thermal stability [5], limiting their applications due to the presence of a sulfate group on its surface [13].

Comment:

As for the application of CNC, the following references are suggested to be added to make a complete literature review: Carbohydr. Polym., 2019, 209, 130-144; ACS Sustain. Chem. Eng., 2020, 8(20), 7536-7562. 

Revision:

Thank you for your suggestion. We added lines 53-61 in the introduction section.

Lines 53-61

Another research has been conducted to produce CNCs with higher thermal stability using the ammonium persulfate (APS) oxidation method. APS oxidation method produces nanocellulose with -COOH surface charge [14]. This method produced CNCs with higher thermal stability than conventional hydrolysis with sulfuric acid [15] [16]. However, compared with other methods (acid hydrolysis and TEMPO oxidation), the APS oxidation method produced a lower surface charge [17].  High thermal stability and high surface charge are specific characteristics of nanocellulose. They were required for expansion in many applications [5] [18] [19] [20] [21] [22] [23]. The surface charge of nanocellulose is described by zeta potential and formed during the isolation or surface modification process.

Reviewer 3 Report

The authors have addressed the issues raised. The paper can be now considered acceptable for publication.

Author Response

Comment:

The authors have addressed the issues raised. The paper can be now considered acceptable for publication.

Revision:

Thank you for helping us improve the quality of our manuscript